# Efficient and Stable Fiber Dye-Sensitized Solar Cells Based on Solid-State Li-TFSI Electrolytes with 4-Oxo-TEMPO Derivatives

**DOI:** 10.3390/nano12132309

**Published:** 2022-07-05

**Authors:** Pyeongje An, Jae Ho Kim, Myeonghwan Shin, Sukyeong Kim, Sungok Cho, Chaehyun Park, Geonguk Kim, Hyung Woo Lee, Jin Woo Choi, Chuljin Ahn, Myungkwan Song

**Affiliations:** 1Department of Energy & Electronic Materials, Korea Institute of Materials Science (KIMS), Changwon 51508, Korea; apj6669@kims.re.kr (P.A.); jho83@kims.re.kr (J.H.K.); chae119@kims.re.kr (S.K.); choso1225@kims.re.kr (S.C.); tlsaudghks98@naver.com (C.P.); rndndndnr2@kims.re.kr (G.K.); 2Department of Nano Fusion Technology, Pusan National University, Busan 46241, Korea; 3Department of Biology and Chemistry, Changwon National University, Changwon 51140, Korea; a463489@kims.re.kr; 4Department of Nanoenergy Engineering and Research Center of Energy Convergence Technology, Pusan National University, Busan 46241, Korea

**Keywords:** fiber-shaped solar cells, dye-sensitized solar cells, solid-state Li-TFSI electrolyte, TEMPO derivatives

## Abstract

Fiber-shaped dye-sensitized solar cells (FDSSCs) with flexibility, weavablity, and wearability have attracted intense scientific interest and development in recent years due to their low cost, simple fabrication, and environmentally friendly operation. Since the Grätzel group used the organic radical 2,2,6,6-tetramethyl-1-piperidinyloxy (TEMPO) as the redox system in dye-sensitized solar cells (DSSCs) in 2008, TEMPO has been utilized as an electrolyte to further improve power conversion efficiency (PCE) of solar cells. Hence, the TEMPO with high catalyst oxidant characteristics was developed as a hybrid solid-state electrolyte having high conductivity and stability structure by being integrated with a lithium bis(trifluoromethanesulfonyl)imide (Li-TFSI) film for FDSSCs. The optimized 4-Oxo TEMPO (OX) based solid-state FDSSC (SS-FDSSC) showed the PCE of up to 6%, which was improved by 34.2% compared to that of the reference device with 4.47%. The OX-enhanced SS-FDSSCs reduced a series resistance (*R*_s_) resulting in effective electron extraction with improved short-circuit current density (*J*_SC_), while increasing a shunt resistance (*R*_sh_) to prevent the recombination of photo-excited electrons. The result is an improvement in a fill factor (FF) and consequently a higher value for the PCE.

## 1. Introduction

Solar energy is a freely available and sustainable source of energy that can be used to meet future energy requirements [1]. Among various solar energy harvesting techniques in flexible devices, fiber-shaped solar cells (FSCs) with a one-dimensional structure that is flexible, wearable, and adaptable to various curved surfaces, such as the human body and textiles, are currently enabling the innovation in present electronic products and reorganizing the future of related fields such as smart textiles [2,3]. Accordingly, various types of FSCs development such as organic solar cells [4,5,6], quantum dot-sensitized solar cells [7,8,9], perovskite solar cells [10,11,12,13,14], and dye-sensitized solar cells [15,16,17,18,19,20,21,22,23] are being attempted and promoted by many researchers. In particular, interest in fiber-shaped dye-sensitized solar cells (FDSSCs), which represent future-oriented flexible and wearable energy sources, is increasing due to their excellent advantages such as low cost, light weight, and simple manufacture. However, since conventional FDSSCs contain an electrolyte in a liquid state, stability of the power conversion efficiency (PCE) of the solar cells is very unstable due to leakage and volatility of the electrolyte. Hence, in our previous studies, solid-state FDSSCs (SS-FDSSCs) were developed with PCE comparable to liquid electrolyte-based devices and excellent stability that were not seen in liquid electrolyte [24,25,26,27,28].

The FDSSCs is generally configured as follows: (i) a transition metal titanium (Ti) wire, which is a core of an anode with high anti-corrosivity and high strength-to-weight ratio resulting in good ductility; (ii) a mesoporous TiO_2_ (mp-TiO_2_) layer as a photocatalysis semiconductor deposited on the anode for electron conduction activation; (iii) dye molecules for charge generation that are covalently bonded as a monolayer to the surface of the TiO_2_ layer to increase absorbance; (iv) an electrolyte containing redox mediator in organic and inorganic solvents effective for dye regeneration; and (v) a platinum (Pt) wire as a cathode with good catalytic properties to facilitate electron collection [29]. Under illumination in the above structure, in the quantum physics theory, incident light excites the electrons at the highest occupied molecular orbital (HOMO) to the lowest unoccupied molecular orbital (LUMO) of the dye. The photo-excited electrons are transferred into the conduction band of the semiconductor metal oxide layer and then moved to the counter electrode through an external circuit. On one hand, iodide (I^−^), a reductive species in the electrolyte, provides electrons to the oxidized dye and becomes triiodide (I_3_^−^), which again obtains electrons from the Pt counter electrode and is reduced to complete the redox couple [30].

The electrolyte is one of the most important components of FDSSCs and has a great influence on the high PCE of the solar cells. The electrolyte in solar cells should have the following properties: (I) It should be possible to efficiently regenerate the oxidized dye. (II) Should not corrode with DSSC components. (III) Should rapidly diffuse the charge carriers, improve conductivity, and enable effective contact between the working electrode and counter electrode. (IV) Absorbance of an electrolyte should not overlap with the absorbance of the dye molecular [31,32]. The recently proposed hybrid electrolyte system can easily improve those electrochemical properties by introducing various additives. Stable organic radicals, such as 2,4,6-tri-*t*-butylphenoxyl, nitronyl nitroxide, and 2,2,6,6-tetramethyl-1-piperidinyloxy (TEMPO), being verified as nontoxic agents, are widely used as antioxidants and light-stabilizers. In particular, since the Grätzel group used TEMPO as an electrolyte for the DSSCs in 2008, numerous efforts have been made to further develop it to increase the PCE of DSSCs [33].

Herein, the TEMPO, an economically and environmentally viable chemical oxidant, was added to the solid-state electrolyte. The fabrication of SS-FDSSCs is described using a novel and stable solid-state electrolyte based on lithium bis(trifluoromethanesulfonyl)imide (Li-TFSI) films impregnated with TEMPO derivatives. The TEMPO-based electrolyte exhibits high transparency compared to commonly used iodine-based commercial electrolyte. In addition, the TEMPO derivative improves the redox reaction in the solid-state electrolyte, thus leading to higher ionic conductivity. The optimized 4-Oxo TEMPO (OX)-enhanced SS-FDSSC boasts effective photon harvest, charge extraction, unidirectional charge transportation, and suppressed charge recombination processes, resulting in a PCE of up to 6%, which was improved by 34.2% compared to that of the reference device with 4.47%. Moreover, the fabricated SS-FDSSCs maintained over 94% efficiency after 500 repetitions of a bending test and over 80% efficiency after 10 repetitions of automatic washing.

## 2. Materials and Methods

### 2.1. Materials

Titanium (Ti) wire of diameter 250 μm (Sigma-Aldrich, St. Louis, MO, USA, 99.7%) was used as the working electrode and platinum (Pt) wire of diameter 127 μm (manufactured domestic company, 99.9%) was used as the counter electrode. Titanium dioxide (TiO_2_) nanoparticle paste (Greatcellsolar, Queanbeyan, NSW, Australia, 18NR-T) was diluted with anhydrous ethanol 99.8% (1:1 *w*/*w*) and stirred with a magnetic bar for 24 h at room temperature. The N719 dye (Greatcellsolar, Ruthenizer 535-bis TBA) was dissolved in anhydrous ethanol 99.8% at a concentration of 0.5 mM. Commercial electrolytes (EL-HSE High Stability Electrolyte and EL-HPE High Performance Electrolyte) were purchased from Greatcellsolar. 4-Oxo-TEMPO (OX) with 95% being purchased from TCI. Using Acetyl chloride (≥99%) purchased from Fluka and Benzenesulfonyl chloride (>99%) and p-Toluenesulfonyl chloride (≥98%) purchased from Aldrich, TEMPO derivatives were produced respectively.

### 2.2. Preparation of the 4-Oxo-TEMPO Derivatives

(a) 3-Acetyl-4-oxo-TEMPO (OAC): To a solution of the 4-Oxo-TEMPO (1 g, 5.87 mmol) and LDA (3.53 mL, 7.04 mmol) in THF is added dropwise acetyl chloride (0.63 mL 8.81 mmol) at 0 °C and the mixture is stirred for 5 h. An ice-cold aqueous 10% hydrochloric acid solution is added to the reaction mixture and the product is extracted with dichloromethane. The combined dichloromethane extract is washed with cold water, dried with magnesium sulfate, and evaporated. After purification by column separation, the product obtained 42% yield.

(b) 3-Benzoyl-4-oxo-TEMPO (OBZ): To a solution of the Oxo-TEMPO (1 g, 5.87 mmol) and LDA (3.53 mL, 7.04 mmol) in THF is added dropwise benzoyl chloride (1.02 mL 8.81 mmol) at 0 °C and the mixture is stirred for 7 h. An ice-cold aqueous 10% hydrochloric acid solution is added to the reaction mixture and the product is extracted with dichloromethane. The combined dichloromethane extract is washed with cold water, dried with magnesium sulfate and evaporated. After purification by column separation, the product obtained 54% yield.

### 2.3. Fabrication of the SS-FDSSCs

After cutting the Ti wire (60 mm), it was washed with deionized water, acetone, and isopropyl alcohol (IPA) in the ultrasonic bath for 10 min. Thereafter, it was dried in a dry oven for 10 min, and electric heating was carried out with a current of 2.0 A for 10 s to form a compact TiO_2_ layer. Electric heating of Ti wires produces compact TiO_2_ layers by oxidation with oxygen in the air. The compact Ti wire thus prepared was dip-coated four times in the diluted TiO_2_ solution, and then annealed at 120 °C for 3 min per dip-coating process. Subsequently, the Ti wire on which the dip-coated TiO_2_ layer was raised was increased from 1.2 A to 2.0 A for 3 min to form a mesoporous TiO_2_ layer by electric heating. The Ti wire adsorbed with a mesoporous TiO_2_ layer was dipped into an N719 dye solution for 20 h. The solid electrolyte Li-TFSI film was wound around the manufactured Ti wire, and then the shape of the solid electrolyte spun with Pt wire was maintained. The prepared device was immersed in an iodine-based electrolyte in which 5 wt% of the TEMPO electrolyte was dissolved for 30 min. The last manufactured device was inserted into a transparent Teflon tube and sealed at both ends with an optical adhesive that can be cured by ultraviolet rays.

### 2.4. Characterization

The optical transmittance was identified via UV-Vis-NIR spectroscopy (Cary 5000, Agilent Technologies, Santa Clara, CA, USA). The photocurrent density–voltage (*J*–*V*) characteristics of the FDSSCs were obtained by an electrometer (Keithley 2400, Cleveland, OH, USA) under air-mass (AM) 1.5 illumination (100 mW cm^−^^2^) provided by a solar simulator (Oriel Sol3A Class AAA solar simulator, models 94043A, Newport, RI, USA). The calibrated light intensity was set to 1 sun using a standard silicon cell. The effective area of the device is defined as the project area transmitted by the mask, which is equal to the diameter of the photoanode multiplied by its length (1 cm). The external quantum efficiency (EQE) was measured using the incident photon-to-current conversion efficiency (IPCE) measurement system (QuantX 300, Newport, Bozeman, MT, USA) with a 250 W quartz-tungsten-halogen lamp, an Oriel Cornerstone^TM^ 130 1/8 m monochromator operated in AC mode, an optical chopper, a lock-in amplifier, and a calibrated Si photodetector. The intensity of the sunlight for outdoor measurements was observed via a UV light meter (TM-208, Tenmars, Neihu, Taipei). Electrochemical impedance spectroscopy (EIS) measurement was measured with an oscillation amplitude of 15 mV under dark conditions (Bio-Logic VMP-3, Seyssinet-Pariset, France) by using the open-circuit voltage and the frequency ranges from 1 Hz to 10 MHz. The experimental data were simulated using commercial Z-view software to estimate the values of each component of the corresponding equivalent circuits. The bending test was performed as a function of the number of bending cycles with a bending radius of 10 mm.

## 3. Results and Discussion

To add carbonyl electrowithdrawing functional groups acting as a radical scavenger to 4-oxo-TEMPO, 4-oxo-TEMPO derivatives such as 3-acetyl-4-oxo-TEMPO (OAC), and 3-Benzoyl-4-oxo-TEMPO (OBZ) were synthesized as shown in Figure 1a. 3-Acetyl-4-oxo-TEMPO (OAC) was produced a 42% yield as crystal solid from the 4-oxo-TEMPO and acetyl chloride with LDA in THF at 0 °C. 3-Benzoyl-4-oxo-TEMPO (OBZ) was produced a 54% yield as crystal solid from the 4-oxo-TEMPO and benzoyl chloride with LDA in THF at 0 °C. OAC was characterized by IR at 1755 cm^−^^1^, 1704 cm^−^^1^ in Figure 1b and OBZ was at 1747 cm^−^^1^, 1725 cm^−^^1^ in Figure 1c, which were the symmetric and asymmetric stretching peaks in the dicarbonyl group.

Various solid-state electrolytes were manufactured by impregnating an iodine-based commercial electrolyte without or with the TEMPO derivatives such as OX, OAC, and OBZ into the Li-TFSI film, respectively. Figure 2a shows the electrical characteristic of the solid-state electrolytes with the various TEMPO derivatives. The high ionic conductivity of the OX-enhanced electrolyte provides the potential for high-performance SS-FDSSCs by causing improved PCE through improved short-circuit current density (*J*_SC_) value due to efficient charge extraction from the electrolyte to the dye. The results indicate that the 4-oxo functional group of the TEMPO derivative is more effective in enhancing the ionic conductivity than either the acetyl-oxo or the benzoyl-oxo group. In detail, it can be interpreted that the electron density of OX decreases by introducing a carbonyl group having an electron-withdrawing effect at position 4 of TEMPO, which reduces the oxidizing power of OX, resulting in an increase in the ionic conductivity of the electrolyte. On the other hand, when comparing OAC having a methyl group on the carbonyl group and the OBZ having a phenyl group on the carbonyl group, in OBZ, it can be interpreted that the electron-withdrawing effect may be offset by conjugating π-electrons of the phenyl group and π-electron of the carbonyl group with each other. Hence, it can be inferred that OAC with the methyl group has an increased electron-withdrawing effect, and its ionic conductivity is lower than that of OBZ, which is relatively small because the electron-withdrawing effect is offset. The UV-Vis transmittance of TEMPO derivatives with various functional groups such as OX, OAC, and OBZ was measured to investigate the optical characteristic, as shown in Figure 2b. The enhanced electrolytes show broad transmittance at about 380 to 500 nm, as shown in inset of Figure 2b. As the absorption rate of OX and OBZ is relatively lower than that of the reference electrolyte, the OX- and OBZ-enhanced electrolytes exhibit higher transmittance than the pristine electrolyte. On the other hand, although OAC has a slightly lower, or almost similar absorption rate to the pristine electrolyte, so the difference is expected to be insignificant. Hence, it was expected that the *J*_SC_ of the SS-FDSSCs with the OX- and OBZ-enhanced electrolytes could be increased as a result of efficient exciton harvesting due to enhanced incident light flex into the dye, whereas the OAC could not.

The photovoltaic performance of TEMPO-based SS-FDSSCs was analyzed by measuring the current density–voltage (*J*–*V*) characteristic curves under AM 1.5 illumination as shown in Figure 3a, and their corresponding photovoltaic parameters were also summarized in Table 1. Conditions of fabrication for SS-FDSSCs in the form of ‘Ti wire/mp-TiO_2_/N719 dye/TEMPO based solid-state electrolyte/Pt wire’ are shown in Appendix A. Four parameters to identify the photovoltaic cell: the *J*_SC_, open-circuit voltage (*V*_OC_), fill factor (FF), and PCE are important factors for analyzing the photovoltaic characteristics of SS-FDSSCs. The OX-enhanced SS-FDSSC demonstrated the highest PCE of 6% (*V*_OC_ = 0.64 V; *J*_SC_ = 13.84 mA cm^−2^; FF = 68.3%), and the OBZ-enhanced SS-FDSSC showed improved PCE of 4.99% (*V*_OC_ = 0.66 V; *J*_SC_ = 10.32 mA cm^−2^; FF = 72.6%), while the pristine electrolyte-based SS-FDSSC had a PCE of 4.47% (*V*_OC_ = 0.65 V; *J*_SC_ = 10.11 mA cm^−2^; FF = 67.6%). Even the OAC-based SS-FDSSC showed the lowest PCE of 4.38% (*V*_OC_ = 0.67 V; *J*_SC_ = 9.25 mA cm^−2^; FF = 70%). It is noteworthy that the OX-enhanced SS-FDSSC with the highest PCE has the lowest *V*_OC_. The theoretical maximum photovoltage of the DSSCs is determined by the energy gap between the redox potential of the electrolyte and the Fermi level of the semiconductor metal oxide on the photoanode, and can only be obtained at zero current. However, output voltage under load is usually less than the *V*_OC_. The voltage loss occurs from the entire overpotential of the counter electrode, which occurs from the current transfer through the electrolyte (mass-transfer overpotential) and through the electrolyte/counter electrode interface (charge-transfer overpotential) [34]. The electron density of OX was decreased by the carbonyl group having the electron-withdrawing effect, thereby decreasing the oxidizing power of OX. As a result, the charge transfer inside the electrolyte was increased, the redox potential of the electrolyte was increased, and the *V*_OC_ would be decreased relatively. As expected, the increased ionic conductivity and high transparency of the electrolytes were related to the enhanced *J*_SC_. The efficient oxidation-reduction at the Pt/electrolyte interface could provide more active charges inside the electrolyte and reduce the charge accumulation. The optimized OX-enhanced SS-FDSSC showed PCE of up to 6%, which was improved by 34.2% compared to that of the reference device with 4.47%.

Dark *J*–*V* characteristics of TEMPO-based SS-FDSSCs were further carried out to analyze the mechanical behavior of the leakage current inside the device, as shown in Appendix A. Reverse saturation current (*J*_0_), which is the current density of the electron-hole recombination, was the lowest in OX in the negative voltage legion. The *J*_0_ were 4.5 × 10^−2^, 4.4 × 10^−2^, 7.7 × 10^−2^, and 4.5 × 10^−2^ mA cm^–2^ for pristine, OX, OAC, and OBZ, respectively, as shown in Appendix A. The OX-based SS-FDSSC show slight inhibition in *J*_0_ compared with the pristine and other TEMPO derivatives, indicating the efficient minimization of the detrimental effect of short circuit pathways [35], leading to an increase in the *R*_sh_, FF, as shown in Table 1. The effect of the TEMPO derivatives on the photocurrent density (*J*_ph_), with the internal voltage (*V*_int_) of the SS-FDSSCs, is shown in Figure 3b. The *J*_ph_, which means a generation degree of photo-excited carriers, was calculated as the difference between the current density under AM 1.5 illumination (Figure 3a) and the current density in the dark condition (Appendix A). The *V*_int_ was calculated as the difference between the built-in voltage when *J*_ph_ was zero and the applied voltage [36]. The calculated *J*_ph_ increased linearly in proportion to the voltage at the low *V*_int_ but was saturated at the relatively high *V*_int_, which can be assumed to be sufficient to sweep all carriers onto the electrode. Thus, a saturated photocurrent density (*J*_ph,sat_) is limited only by the number of absorbed photons. The values of *J*_ph,sat_ obtained from the highest values of *J*_ph_ were 10.1, 13.8, 9.2, and 10.3 mA cm^−2^, in order of the pristine, OX, OAC, and OBZ, respectively. The charge collection and blocking behavior of the TEMPO-based SS-FDSSCs were investigated to analyze the effect on the electrical properties of TEMPO-based electrolyte, as shown in Appendix A. The charge collection probability (*P*_cc_) with respect to *V*_int_ under AM 1.5 illumination can be obtained by the *J*_ph_ and *J*_ph,sat_ of the solar cells. The *P*_cc_ can be obtained by normalizing *J*_ph_, which is divided by *J*_ph,sat_ [37]. As a result, the *P*_cc_ on OX-enhanced SS-FDSSC was shown to be slightly higher than that on the pristine electrolyte based SS-FDSSC in the full range from short-circuit to open-circuit conditions.

Electrochemical impedance spectroscopy (EIS) was carried out under dark conditions in a frequency range of 0.1 Hz ~ 1 MHz at a bias voltage close to *V*_OC_ (0.7 V) to better understand the charge transfer dynamics in the TEMPO-based electrolyte, and the results are shown in the Nyquist plots in Figure 3c. The Nyquist plot, which can evaluate the interfacial characteristics of the TEMPO-based SS-FDSSCs, was fitted using an equivalent circuit diagram, as shown in Appendix A. The device based on OX showed the *R*_s_ of 15.9 Ω, which is smaller than the corresponding values of OAC (17.2 Ω) and OBZ (19.9 Ω). The pristine electrolyte-based device (25.0 Ω) even showed a large *R*_s_ value, as shown in Appendix A. The *R*_1_ and *R*_2_ values were 41.5 and 116 Ω (pristine electrolyte), 41.9 and 104 Ω (OX), 48.1 and 144 Ω (OAC) and 39.3 and 107 Ω (OBZ), respectively. The lowest *R*_s_ value was obtained from OX-based SS-FDSSC, which means that the OX-enhanced electrolyte enhances the charge transfer at the counter electrode/electrolyte because of its higher ionic conductivity. Based on the improved charge transfer characteristics at the electrolyte, the key parameters of the SS-FDSSCs could be enhanced. Therefore, the improved *J*_SC_ and FF values enhance the PCE of the SS-FDSSCs.

In order to further analyze the influence of OX on the electrical characteristics of the SS-FDSSC as shown in Figure 3d, a sublinear dependence of *J*_SC_ on light intensity (*I*) according to the power law [38], which *J*_SC_ ∝ *I^α^*, where *α* < 1 is the exponential factor. The fact that *J*_SC_ is nearly linear means that the losses of charge-carrier in the absorber were dominated by monomolecular recombination, and it can be confirmed that pure bimolecular recombination was not observed even at the highest illumination intensity [39]. The data were represented on a logarithmic scale on both the *x* and *y* axes and fitted to the power law: *α* = 0.992 and *α* = 0.946 for the OX and pristine, respectively, indicating that bimolecular recombination was decreased in the OX-enhanced SS-FDSSC. This result is consistent with the overall increase in *P*_cc_ under short-circuit conditions, as well as the increase in FF for all OX-enhanced devices. The *J*–*V* curves of the OX-enhanced SS-FDSSC under 0.2 to 1.0 sun illumination were obtained as shown in Appendix A. As the light intensity increased, the *J*_SC_ and *V*_OC_ were seen to increase. These results indicate that as the photo-excited electrons of the dye moved to the semiconductor metal oxide layer, the Fermi level of the semiconductor metal oxide layer increased and thus *J*_SC_ and *V*_OC_ increased [40,41].

The SS-FDSSC, which uses Ox-enhanced electrolyte, showed a remarkable degree of flexibility and durability, as shown in Figure 4a and Appendix A. The PCE of the OX-enhanced SS-FDSSC, with a radius of approximately 5 mm, changed slightly to almost 94% after 500 bending cycles. This is because wearable electronics must be both flexible and washable. Washing tests of OX-enhanced SS-FDSSC were carried out in an automatic washer, as shown in Figure 4b and Appendix A. During the 10 repeated washing tests, the device stability tests showed a little degradation in the device performance. These results indicate that OX-enhanced SS-FDSSCs have excellent characteristics as wearable electronic devices. The stability for *J*_SC_ and *V*_OC_ were also presented to provide more information on the stability test for bending and washing, which can be seen in Appendix A. An aging test was carried out to assess the stability of the fabricated OX-enhanced SS-FDSSCs under indoor ambient conditions, as shown in Figure 4c. The devices exhibited reasonable stability for 14 days. Durability of OX-enhanced SS-FDSSC against extreme environments was tested to high temperature up to 200 °C, as shown in Figure 4d. The PCE of OX-enhanced SS-FDSSCs maintained an efficiency of 93% compared to the initial efficiency even in hot environments with increasing temperature.

## 4. Conclusions

In conclusion, the TEMPO derivatives and chemical oxidant were added to the solid-state electrolyte. The TEMPO-based electrolyte exhibits high transparency compared to commonly used iodine-based commercial electrolyte. In addition, the TEMPO derivative improves the redox reaction in the solid-state electrolyte, thus leading to higher ionic conductivity. The optimized OX-enhanced SS-FDSSC boasts effective photon harvest, charge extraction, unidirectional charge transportation, and suppressed charge recombination processes, resulting in a PCE of up to 6%, which was improved by 34.2% compared to that of the reference device with 4.47%. Moreover, to demonstrate the improvement in *J*_SC_, which resulted in improved PCE due to OX, the optical properties of the devices were analyzed to confirm the increase in *J*_ph_, which resulted in an increase in *P*_cc_. The experimental approach and outcomes of this study present an efficient solution that can improve the performance of SS-FDSSCs using OX as the electrolyte additive.

## Figures and Tables

**Figure 1 nanomaterials-12-02309-f001:**
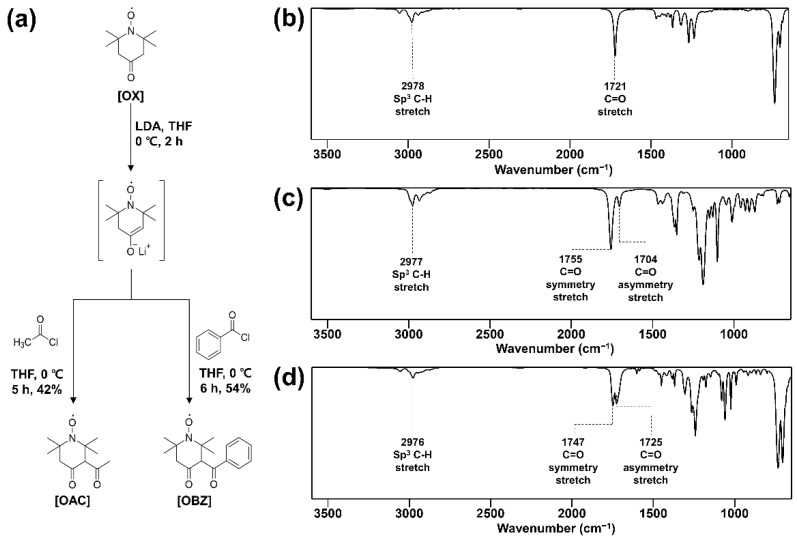
Characteristics of TEMPO derivatives: (**a**) synthetic scheme with molecular structures, and infrared spectrum of (**b**) OX, (**c**) OAC, (**d**) OBZ.

**Figure 2 nanomaterials-12-02309-f002:**
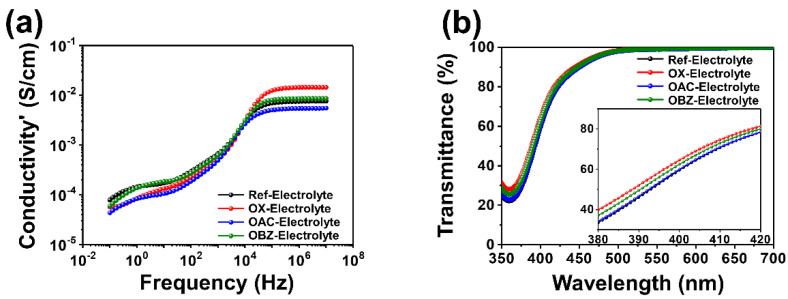
Characteristics of TEMPO derivatives: (**a**) ionic conductivities when absorbed onto the Li-TFSI film for use as a solid-state electrolyte, and (**b**) transmittance when diluted onto the ethanol.

**Figure 3 nanomaterials-12-02309-f003:**
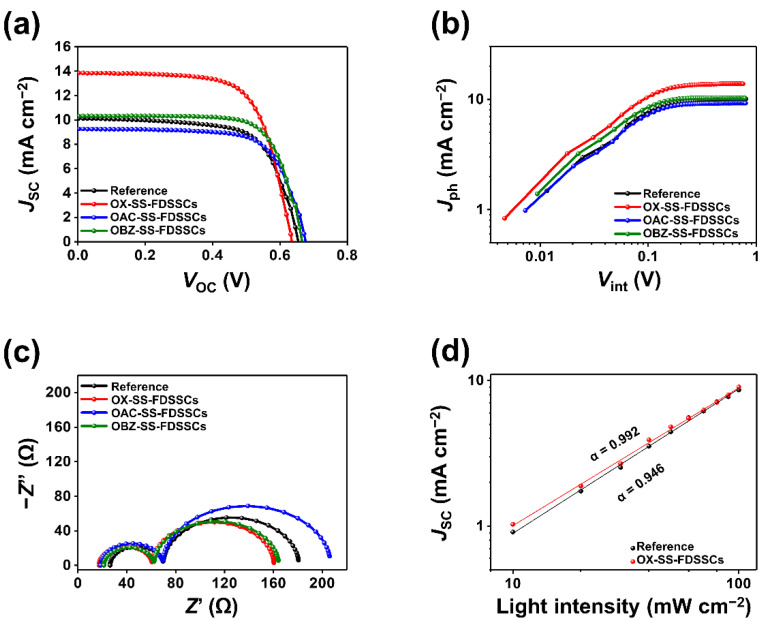
Characterization and photovoltaic properties of TEMPO-SS-FDSSCs devices: (**a**) *J*–*V* curves, (**b**) photocurrent density as a function of internal voltage, and (**c**) nyquist plots measured by EIS data. (**d**) Measured *J*_SC_ of OX-enhanced SS-FDSSC plotted against light intensity (symbols) on a logarithmic scale. Fitting a power law (solid lines) to these data yields α.

**Figure 4 nanomaterials-12-02309-f004:**
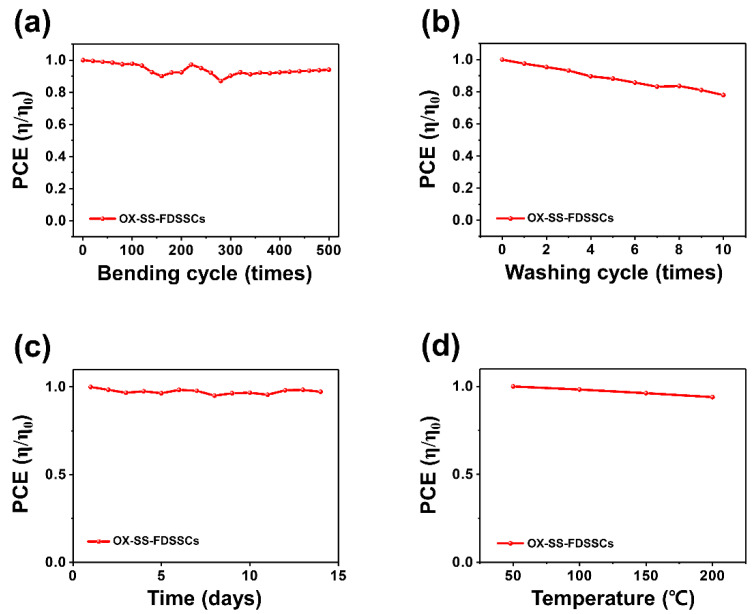
Normalized PCE of SS-FDSSCs with TEMPO derivative (OX) as a function of (**a**) bending cycle and (**b**) washing cycle. (**c**) Stabilities under indoor ambient condition, and (**d**) heating condition in dry oven.

**Table 1 nanomaterials-12-02309-t001:** Photovoltaic properties of SS-FDSSCs as a function of TEMPO derivatives.

	*V*_OC_ (V)	*J*_SC_ (mA cm^−2^)	FF (%)	PCE (%)	*R*_s_ (Ω cm^2^)	*R*_sh_ (Ω cm^2^)
Pristine	0.65	10.11	67.6	4.47	7.20	1.29 × 10^3^
OX	0.64	13.84	68.3	6.00	6.15	7.06 × 10^3^
OAC	0.67	9.25	70.0	4.38	8.08	1.25 × 10^3^
OBZ	0.66	10.32	72.6	4.99	6.88	2.55 × 10^3^

## Data Availability

Not applicable.

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
