# Peer review of "Efficient and Stable Fiber Dye-Sensitized Solar Cells Based on Solid-State Li-TFSI Electrolytes with 4-Oxo-TEMPO Derivatives"

_nanomaterials, 2022, doi:10.3390/nano12132309_

Round 1

Reviewer 1 Report

The paper titled “Efficient and stable fiber dye-sensitized solar cells based on solid-state Li-TFSI electrolytes with -4Oxo-TEMPO derivatives” was submitted in Nanomaterials. I recommend the article for publication after addressing the following major comments.  

1.     The slope in Figure 3d is higher than 1 for both cases. Usually, it should be between 0.75 to 1. Please explain.  

2.     What is the significance of Figure 3b? At what values of Vin, Jph got saturated? There is no clarity. Also, it is suggested to provide dark curves of devices.

3.     There is further explanation required for the significantly low VOC of the OX device (Table 1)? Also, low Rsh is not desirable for indoor lights because it is associated with leaking current. How to fix it?

4.     The authors have performed stability and washing tests. However, there is no concrete analysis apart from PCE, provided for the improved performance of the OX-SS-FDSSC device. It seems ambiguous without further characterizations. Also, can bend the devices < 5mm?

What is the importance of this study for indoor light energy harvesting as VOC is also low? Please comment and include the related studies in the introduction.   (https://doi.org/10.1016/j.dyepig.2021.109624; https://doi.org/10.1016/j.dyepig.2021.109626).  

Author Response

Thank you for handling our manuscript entitled “Efficient and stable fiber dye-sensitized solar cells based on solid-state Li-TFSI electrolytes with 4-Oxo-TEMPO derivatives” (Manuscript ID: nanomaterials-1784828)

We have carefully read the reviewers’ comments describing our work and observations as original, timely, and important. We have revised our manuscript in accordance with the reviewers’ comments. These corrections are highlighted yellow for first reviewer and highlighted green for second reviewer in the revised manuscript. We believe that we have satisfactorily addressed the reviewers’ technical concerns our revised manuscript.

With the incorporated changes, we hope the revised manuscript is now acceptable for publication in “Nanomaterials”. Should you have any further questions or comments, please let me know.

Detailed responses to the reviewers’ comments are enumerated below:

Reviewer 2 Report

An and Kim et al. present a 4-oxo TEMPO as an additive for the electrolyte to optimize solid-state Fiber-shaped dye-sensitized solar cells (SS-FDSSC). The device shows a huge increase in JSC and the final PCE is 6.00% which is improved by 34.2% compared to the reference device. Also, the durability is proved enhanced after bending and washing tests. It’s a very fantastic material and method. Therefore, I recommend it publish in nanomaterials after addressing the following issues with a major revision.

1.    “Pcc” in line 255 and “JSC” in line 276 did not use subscript; the “providres” in line 175 is wrongly spelled.

2.    The description of figure 1(b,c,d) in the text does not correspond to the correct figures.

3.    The article said “The enhanced electrolyte shows a broad transmittance at 300 to 500 nm with a transmission peak centered at around 325 nm”, but the corresponding figure starts from 350 nm. Can you also include the data from 300 to 350 nm?

4.    It should be stated that whether the various durability tests were operated under illumination or dark environment.

5.    The enhanced FF of OAC and OBZ devices should be explained.

6.    The NMR (1H and 13C) spectrums of OAC and OBZ should be provided.

Author Response

(The authors gave the same response as above.)

Round 2

Reviewer 1 Report

No more revisions are required

Reviewer 2 Report

It could be accepted now.